

# Transcriptome dynamics along axolotl regenerative development are consistent with an extensive reduction in gene expression heterogeneity in dedifferentiated cells

Carlos Díaz-Castillo

Irvine, CA, USA

Corresponding author
Carlos Díaz-Castillo,
crlsdiazcastillo@gmail.com

## ABSTRACT

Although in recent years the study of gene expression variation in the absence of genetic or environmental cues or gene expression heterogeneity has intensified considerably, many basic and applied biological fields still remain unaware of how useful the study of gene expression heterogeneity patterns might be for the characterization of biological systems and/or processes. Largely based on the modulator effect chromatin compaction has for gene expression heterogeneity and the extensive changes in chromatin compaction known to occur for specialized cells that are naturally or artificially induced to revert to less specialized states or dedifferentiate, I recently hypothesized that processes that concur with cell dedifferentiation would show an extensive reduction in gene expression heterogeneity. The confirmation of the existence of such trend could be of wide interest because of the biomedical and biotechnological relevance of cell dedifferentiation-based processes, i.e., regenerative development, cancer, human induced pluripotent stem cells, or plant somatic embryogenesis. Here, I report the first empirical evidence consistent with the existence of an extensive reduction in gene expression heterogeneity for processes that concur with cell dedifferentiation by analyzing transcriptome dynamics along forearm regenerative development in *Ambystoma mexicanum* or axolotl. Also, I briefly discuss on the utility of the study of gene expression heterogeneity dynamics might have for the characterization of cell dedifferentiation-based processes, and the engineering of tools that afforded better monitoring and modulating such processes. Finally, I reflect on how a transitional reduction in gene expression heterogeneity for dedifferentiated cells can promote a long-term increase in phenotypic heterogeneity following cell dedifferentiation with potential adverse effects for biomedical and biotechnological applications.

# INTRODUCTION

One of the most intriguing aspects of biological systems is that they are prone to vary even in the absence of genetic or environmental cues (*Ackermann, 2015*; *Altschuler & Wu, 2010*; *Komin & Skupin, 2017*; *Liu, Francois & Capp, 2016*; *Symmons & Raj, 2016*). Non-genetic,

non-environmental phenotypic heterogeneity is dependent on the inherent stochasticity of the physicochemical substrate of biological systems, but also on emergent properties of biological processes like: (i) the frequent dependence of biological processes on low numbers of intervening elements, (ii) the crowded conditions of intracellular compartments, (iii) the fidelity decrease in the multistep transference of information to *de novo* synthesize active genetic products, (iv) the pulsating nature of RNA transcription, or, (v) the slow dynamics of highly compacted chromatin remodeling needed to grant DNA accessibility to the transcription machinery (*Ackermann, 2015*; *Altschuler & Wu, 2010*; *Komin & Skupin, 2017*; *Liu, Francois & Capp, 2016*; *Rivas & Minton, 2016*; *Symmons & Raj, 2016*; *Yanagida et al., 2015*). Since some of these factors can vary within single cells temporally or in response to environmental cues, between cells within single multicellular organisms, or between unicellular or multicellular organisms within single populations, it would be expected that non-genetic non-environmental phenotypic heterogeneity was itself variable (*Ackermann, 2015*; *Altschuler & Wu, 2010*; *Komin & Skupin, 2017*; *Liu, Francois & Capp, 2016*; *Rivas & Minton, 2016*; *Symmons & Raj, 2016*; *Yanagida et al., 2015*). Thus, the study of variation patterns for non-genetic non-environmental heterogeneities is of potential interest for the characterization of biological properties that promoted such heterogeneities.

Recently, Díaz-Castillo proposed that processes that concur with cell dedifferentiation would show an extensive reduction in gene expression heterogeneity (*Díaz-Castillo, 2017b*), i.e., the variation in gene expression detected between cells or organisms with the same genotype when assayed in the same environmental conditions. Cell dedifferentiation refers to cases in which well-differentiated, specialized, non-proliferative cells revert to states characterized by less specialization, the ability of re-differentiating towards different cellular fates, and/or proliferation (*Li & Belmonte, 2017*; *Merrell & Stanger, 2016*; *Sugiyama, 2015*; *Yamada, Haga & Yamada, 2014*). Cell dedifferentiation is known to occur in the initial stages of developmental programs activated in response to injury or regenerative development in vertebrates, the formation of masses of undifferentiated cells such as tumors in animals or *calli* in plants, or the artificial induction of somatic embryogenesis in plants and human pluripotent stem cells for biomedical applications (*Li & Belmonte, 2017*; *Merrell & Stanger, 2016*; *Sugiyama, 2015*; *Yamada, Haga & Yamada, 2014*).

The proposal that an extensive decrease in gene expression heterogeneity is characteristic of processes based on cell dedifferentiation relies on cell dedifferentiation itself being a case of cell convergence, and on dedifferentiating cells tendency to show an extensive relaxation of chromatin (*Díaz-Castillo, 2017b*). Since cell dedifferentiation represents the reversal of cell specialization, it would be expected that dedifferentiated cells were more similar between them than the specialized cells they originated from are. Indeed, the use of information theory to characterize transcriptomes for diverse types of human and murine tumors and the organs they originated from showed not only that cancer transcriptomes become less specialized and more similar among themselves than the transcriptomes of their original organs are, but also that they become very similar to the transcriptomes of undifferentiated embryonic stem cells (*Martinez & Reyes-Valdes, 2008*; *Martinez, Reyes-Valdes & Herrera-Estrella, 2010*).

On the other hand, based on the analysis of chromatin markers and the activation of genetic elements usually silent because of their location in chromosome regions with highly compacted chromatin, it has been suggested that both naturally-occurring and artificially-induced cell dedifferentiation is characterized by an extensive relaxation of chromatin throughout the nucleus (*El-Badawy & El-Badri, 2015*; *Feher, 2015*; *Grafi & Barak, 2015*; *Jiang, Zhu & Liu, 2013*; *Krause, Sancho-Martinez & Izpisua Belmonte, 2015*; *Lee et al., 2015*; *Macia, Blanco-Jimenez & Garcia-Perez, 2015*; *Sosnik et al., 2017*; *Wang & Wang, 2012*; *Zhu et al., 2012a*). Ultimately, chromosome regions with high chromatin compaction are supposed to promote gene expression heterogeneity because of the slow dynamics of the chromatin remodeling needed to grant DNA accessibility to the transcription machinery (*Liu, Francois & Capp, 2016*; *Symmons & Raj, 2016*). The modulating effect chromatin compaction has on gene expression heterogeneity is best exemplified by the phenomenon known as position effect variegation (PEV), and the effect of chromatin compaction modifiers on gene expression and phenotypic heterogeneities. Originally discovered in *Drosophila melanogaster*, PEV refers to the stochastic variation in expression for genes located close to or embedded within chromosome regions with highly compacted chromatin or heterochromatin (*Elgin & Reuter, 2013*). Factors that directly or indirectly alter chromatin compaction such as temperature, genetic variation in heterochromatin-forming elements, paternal/maternal chromosome inheritance, or the genomic content in junk DNA have been shown to modulate gene expression heterogeneity itself and the phenotypic manifestation of such heterogeneity as PEV (*Díaz-Castillo, 2015*; *Elgin & Reuter, 2013*; *Maggert & Golic, 2002*). Thus, considering that chromatin compaction promotes gene expression heterogeneity, it could be inferred that the extensive reduction in chromatin compaction in dedifferentiating nuclei would cause a reduction of gene expression heterogeneity for many genes throughout the genome.

Because no formal proof exists yet for the extensive reduction in gene expression heterogeneity in dedifferentiating cells, I aimed to find preliminary supporting evidence by focusing on the study of gene expression dynamics along regenerative development in *Ambystoma mexicanum* or axolotl. First, although not yet completely understood, cell dedifferentiation associated to axolotl regenerative development in response to injury has been and still is an important subject of study (*McCusker, Bryant & Gardiner, 2015a*). Second, transcriptomic analyses of regenerative development in axolotl constitute an ideal model to study gene expression dynamics for processes that concur with cell dedifferentiation because it would permit studying naturally occurring cell dedifferentiation-based phenomena from the moment they are elicited until their completion. Finally, recent years have seen an increase in transcriptomic studies for axolotl regenerative development models (*Bryant et al., 2017*; *Eo et al., 2012*; *Gearhart et al., 2015*; *King & Yin, 2016*; *Knapp et al., 2013*; *McCusker et al., 2015b*; *Monaghan et al., 2012*; *Monaghan et al., 2009*; *Pai et al., 2016*; *Ponomareva et al., 2015*; *Sabin et al., 2015*; *Seifert et al., 2012*; *Sousounis et al., 2014*; *Stewart et al., 2013*; *Voss et al., 2015*; *Wu et al., 2013*). Here, I argue about the adequacy of using one these studies to test the prediction for gene expression heterogeneity dynamics associated to cell dedifferentiation because of its temporal design and remarkable replication level. The results of reanalyzing the
chosen dataset are consistent with the possibility that processes that concur with cell dedifferentiation might be characterized by an extensive reduction in gene expression heterogeneity (*Díaz-Castillo, 2017b*). In addition, I briefly reflect on the usefulness of the study of gene expression heterogeneity patterns associated to cell dedifferentiation for biomedical and biotechnological applications.

## MATERIALS AND METHODS

Data from the transcriptome study for axolotl regenerative development with the largest biological replication was used to assess if lower levels of biological replication, 3–6 replicates, could limit the study of gene expression heterogeneity patterns along axolotl regenerative development (*Bryant et al., 2017*; *Eo et al., 2012*; *Gearhart et al., 2015*; *King & Yin, 2016*; *Knapp et al., 2013*; *McCusker et al., 2015b*; *Monaghan et al., 2012*; *Monaghan et al., 2009*; *Pai et al., 2016*; *Ponomareva et al., 2015*; *Sabin et al., 2015*; *Seifert et al., 2012*; *Sousounis et al., 2014*; *Stewart et al., 2013*; *Voss et al., 2015*; *Wu et al., 2013*). In 2015, Voss and coworkers inspected gene expression dynamics for 9–10 biological replicates at 20 timepoints along 28 days following axolotl forearm amputation using custom Affymetrix GeneChips that include 20,080 probesets and RNA samples obtained from 1 mm of heterogeneous tissue from the tips of amputated forearms (*Voss et al., 2015*). The dataset including normalized transcript abundance for this study, Voss dataset hereinafter, was obtained from Gene Expression Omnibus database (GSE67118) (*Barrett et al., 2013*; *Voss et al., 2015*). Timepoint, biological replicate, and probeset identifiers from the original study were maintained here (*Voss et al., 2015*).

For each probeset and timepoint, transcript abundance mean and coefficient of variation (CV) were calculated and used as proxies for gene expression level and heterogeneity respectively (Dataset S1). Next, for each probeset and timepoint, transcript abundance and CV using three, four, five, or six biological replicates chosen at random were calculated to simulate the effect lower replication would have for gene expression measures reproducibility. The similarity of transcript abundance mean and CV for all probesets in each timepoint in the Voss dataset when calculated using all biological replicates and each lower-replication simulation was inspected using Pearson *r*. Lower-replication random simulations were repeated 1,000 times. Table S1 summarizes Pearson *r* for each timepoint, while Fig. S1 represents Pearson *r* calculated for three timepoints chosen at random. Increasing biological replication results in a similarity improvement for both transcript abundance measures when compared to measures calculated using all biological replicates. However, the similarity of gene expression measures using lower-replication simulations and all biological replicates is always considerably worse for transcript abundance CV than for transcript abundance mean. Thus, although low biological replication can still be appropriate to study changes in the level of gene expression in response to natural or experimental variables, the study of gene expression heterogeneity dynamics warrants the use of designs with abundant biological replication. For this reason, in subsequent analyses I proceed to use only the Voss dataset, which ensures the largest biological replication level for transcriptomic studies of axolotl regenerative development.

The Monte Carlo-Wilcoxon matched-pairs signed-ranks test (MCW test hereinafter) was used to study gene expression temporal dynamics along the regenerative development of axolotl forearm using the Voss dataset as reference (*Díaz-Castillo, 2015*; *Voss et al., 2015*). In essence, MCW is based on the Wilcoxon matched-pairs signed-ranks test which permits assessing if matched pairs of quantitative data sampled in two different conditions tend to be biased in any particular direction (*Wilcoxon, 1945*). This test proceeds by calculating a sum of signed ranks assigned to the elements in the dataset in question *in virtue* of the subtraction of their matched data. The sum of signed ranks ($W$) is sensitive both to the number of elements with data biased in one or the other direction, and the extent of these biases. The MCW test variation evaluates the significance of the observed bias for the dataset under study by comparing $W$ calculated before and after randomly rearranging data. MCW is very versatile because data can be randomly rearranged with no restriction or respecting certain aspects of the internal structure of the dataset to simulate the effect different factors might have on the variation of the measure under study. Here, MCW tests were used to asses if transcript abundance CV was generally lower in post-amputational timepoints (TX) than in the day of the amputation (T0), TX vs T0 comparisons hereinafter. For each TX vs T0 comparison, I calculated a gene expression bias index using transcript abundance CV for each probeset in the dataset (cvGEBI). The main steps for the calculation of cvGEBI are represented in Fig. S2. First, for each probeset in the dataset, I subtracted transcript abundance CV for the post-amputational timepoint from transcript abundance CV for the day of the amputation ($\Delta \text{CV} = \text{CV}_{\text{TX}} - \text{CV}_{\text{T0}}$). Second, probesets were ranked after sorting them using the absolute value of $\Delta \text{CV}$ from smallest to largest. Third, ranks for each probeset were signed using the sign of $\Delta \text{CV}$. Fourth, the sum of signed ranks ($W$) was calculated as $\sum [\text{sgn}(\Delta \text{CV}_i) \cdot rk(|\Delta \text{CV}_i|)]$. Fifth, cvGEBI was calculated as $W / W_{\text{max}}$, where $W_{\text{max}}$ represents the maximum value $W$ could take. cvGEBIs will range from 1 to $-1$ if transcript abundance CV was higher or lower post-amputationally than in the day of the amputation for all the probesets in the dataset, respectively.

MCW tests evaluate the significance of observed cvGEBIs by comparing them with simulated cvGEBIs calculated after randomly rearranging transcript abundance CV. Here, three different MCW test designs were used to simulate the effect chance, factors acting on the transcriptome as a whole, or the variation in the mean level of expression would have on transcript abundance CV dynamics along axolotl forearm regenerative development. These three MCW designs are referred to as unrestricted, timepoint-restricted, and expression-restricted MCW tests (Fig. S2, and Table S2). Unrestricted MCW tests proceed by recalculating cvGEBIs after randomly rearranging transcript abundance CV with no other restriction. Timepoint-restricted MCWs tests proceed by recalculating cvGEBIs after randomly rearranging transcript abundance CV within each timepoint for each TX vs T0 comparison. Expression-restricted MCW tests proceed by recalculating cvGEBIs after randomly rearranging transcript abundance CV within bins of probesets defined by their corresponding transcript abundance mean values. Probeset bins were defined by rounding up transcript abundance mean values to two or three decimal digits independently. These two alternatives differ in the numbers of bins and number of probesets per bin (Table S3). Random permutations of transcript abundance CV for each MCW test were repeated

10,000 times. The significance of observed cvGEBIs was estimated by calculating $P_{upper}$ and $P_{lower}$ values as the fraction of simulated cvGEBIs that were higher or equal, or lower or equal than observed cvGEBIs, respectively. Observed cvGEBIs were considered significant if $P_{upper}$ and $P_{lower}$ values were lower than 0.05.

The proportional cumulative area under the curve ($PC_{AUC}$) was used to estimate how much of the temporal dynamics for gene expression heterogeneity along the regenerative development of axolotl forearm could be accounted exclusively by chance, factors acting on the transcriptome as a whole, or the variation in the mean level of expression for each probeset (Table S4). $PC_{AUC}$ was calculated as the sum of simulated cvGEBIs with the closets value to observed cvGEBIs for the corresponding TX *vs* T0 comparison and all precedent ones divided by the sum of observed cvGEBIs for the corresponding TX *vs* T0 comparison and all precedent ones for unrestricted, timepoint-restricted and expression-restricted MCW tests.

Unrestricted MCW tests were also used to study the dynamics of the mean level of gene expression along axolotl forearm regenerative development (Table S2). In this case, gene expression bias indexes for each TX vs T0 comparison were calculated following the process previously described but using transcript abundance mean for each probeset and timepoint, and referred to as mGEBIs. The statistical significance of observed mGEBIs was inspected as previously described for cvGEBIs.

To functionally characterize those genes that contribute to the sharp decrease in gene expression heterogeneity early in the regenerative development of axolotl forearm, gene expression heterogeneity fold change between 1 and 1.5 days post-amputation was calculated for each probeset in the dataset as $\log_2(CV_{T1.5}/CV_{T1})$, where, $CV_{T1}$ and $CVT_{1.5}$ correspond to transcript abundance CV calculated for T1 and T1.5 timepoints, respectively. In an attempt to distinguish probesets for which gene expression heterogeneity dynamics between 1 and 1.5 days post-amputation constituted a clear tendency change from those that showed erratic fluctuations within a longer timeline, transcript abundance CV data for 0.5 and 2 days post-amputation were also taken into consideration. For those probesets for which transcript abundance CV was not higher or lower for both 0.5 and 1 days timepoints than for both 1.5 and 2 days timepoints, their $\log_2(CV_{T1.5}/CV_{T1})$ was set to 0. GOrilla and REVIGO were used to identify significant enrichments for Gene Ontology (GO) terms (*Eden et al., 2009*; *Supek et al., 2011*). GOrilla permits detecting GO term enrichments for genes in the top of a ranked list when compared with the rest of the list using a minimum hypergeometric (mHG) test (*Eden et al., 2009*). GOrilla was used to find GO enrichments for all probesets in the dataset ranked using the increasing value of corrected $\log_2(CV_{T1.5}/CV_{T1})$, and 10 lists including all probesets in the dataset randomly rearranged setting the reference GO annotation to *Homo sapiens*, the running mode to "single ranked list of genes", and, the $P$ value threshold to $10^{-3}$ (Table S5). The GO term enrichment found using randomly rearranged probesets supported by the largest number of genes was used as the threshold to narrow down observed GO enrichments with potential biological significance. REVIGO was used to minimize GO term redundancies

using *P* values obtained using GOrilla as GO enrichments, and setting allowed similarity to medium, the database with GO term sizes to whole UniProt, and, the semantic similarity to SimRel.

The MCW test was also used to study gene expression dynamics for specific subsets of genes defined by their similar functionality when compared with the whole transcriptome along axolotl forearm regenerative development (Fig. S2, and Table S2). This new MCW variation is referred to as functionally-restricted MCW test (Fig. S2). First, genes classified in human gene ontology classes mitotic cell cycle process (GO:1903047), and chromatin organization (GO:0006325) were retrieved from the Gene Ontology Consortium database *The Gene Ontology Consortium, 2015*). Functionally-restricted MCWs tests proceed by calculating mGEBIs and cvGEBIs for groups of probesets corresponding to genes classified in each GO class before and after randomly rearranging functional subset tags for the whole dataset and each TX *vs* T0 comparison 10,000 times. The statistical significance of observed mGEBIs and cvGEBIs using functionally-restricted MCW tests was inspected as previously described.

## RESULTS

### An extensive reduction in gene expression heterogeneity is detectable early after axolotl forearm amputation

The axolotl has been a preferred model system for the study of post-traumatic regeneration of complex structures for more than a century (*Voss, Woodcock & Zambrano, 2015*). During this time, the progress of regenerative development responses has been intensively characterized macroscopically and microscopically (*McCusker, Bryant & Gardiner, 2015a*). Briefly, immediately after a gross insult such as the amputation of a limb, a number of processes are activated in response to the injury to heal the open wound (*McCusker, Bryant & Gardiner, 2015a*). Shortly after that, the amputation plane gets populated with regeneration-competent progenitor cells, collectively referred to as the blastema, which will coordinately grow and pattern to restore the missing structure (*McCusker, Bryant & Gardiner, 2015a*). A long-held view assumes that dedifferentiating cells are the main contributor to the formation of the blastema, and evidence of cell dedifferentiation upon injury have been found for different cell types such as muscle cells, keratinocytes or fibroblasts (*McCusker, Bryant & Gardiner, 2015a*). However, recent lineage-tracing analyses raised some doubts on how many cell types actually dedifferentiate in response to injury, and for those that do, how they contribute to the regeneration of the different tissues needed to restore the missing structure (*Kragl et al., 2009*; *Sandoval-Guzman et al., 2014*; *Wu et al., 2015*). Conversely, the study of heterogeneous samples from post-amputation axolotl limb blastemas have shown the activation shortly after amputation of genes commonly expressed in germ line cells, and transposable elements (TE) commonly silent because of their location in regions with highly compacted chromatin (*Zhu et al., 2012a*; *Zhu et al., 2012b*). Since TE activation is known to occur as a consequence of the extensive chromatin relaxation in un/dedifferentiated cells (*Feher, 2015*; *Grafi & Barak, 2015*; *Macia, Blanco-Jimenez & Garcia-Perez, 2015*; *Wang & Wang, 2012*), TE activations detected in axolotl regenerative

development responses might be consistent with the possibility that the formation of the blastema required, at least partially, dedifferentiating cells that undergo extensive chromatin relaxation. Furthermore, *Sosnik et al. (2017)* recently developed a new method to quantify the nuclear condensation using microscopic images and demonstrated that blastemal cells show a significant reduction in chromatin condensation when compared with somatic dermal cells. Thus, the use of transcriptomic analyses for axolotl regenerative development models seems like an adequate place to seek preliminary support for the hypothesized extensive decrease in gene expression heterogeneity for processes that concur with cell dedifferentiation (*Díaz-Castillo, 2017b*).

In recent years, the number of studies of transcriptome dynamics for axolotl regenerative development models increased considerably (*Bryant et al., 2017*; *Eo et al., 2012*; *Gearhart et al., 2015*; *King & Yin, 2016*; *Knapp et al., 2013*; *McCusker et al., 2015b*; *Monaghan et al., 2012*; *Monaghan et al., 2009*; *Pai et al., 2016*; *Ponomareva et al., 2015*; *Sabin et al., 2015*; *Seifert et al., 2012*; *Sousounis et al., 2014*; *Stewart et al., 2013*; *Voss et al., 2015*; *Wu et al., 2013*). A common aspect to many of these studies is that they encompass very low levels of biological replication, i.e., 1–6 replicates per condition, which can constitute a big obstacle for the study of gene expression heterogeneity dynamics (see Materials and Methods, Fig. S1, and Table S1). For this reason, all analyses presented here used the dataset produced by the transcriptome study for axolotl regenerative development with the largest biological replication level. In 2015, Voss and coworkers analyzed transcriptome dynamics during the initial 28 days of axolotl post-amputation forearm regenerative development using 1 mm of heterogeneous tissue from the tip of amputated and post-amputational regenerating forearms at 19 timepoints (*Voss et al., 2015*). More importantly, the study performed by Voss and coworkers consisted of 9–10 biological replicates per timepoint for a region of post-amputational regenerating forearms where the blastema is formed and maintained along the regenerative process (*Voss et al., 2015*).

To characterize gene expression heterogeneity dynamics along axolotl forearm regenerative development, transcript abundance coefficient of variation (CV) was used as a measure of gene expression heterogeneity for each probeset and timepoint in Voss dataset. Also, a variation of the Monte Carlo-Wilcoxon matched-pairs signed-ranks test (MCW test hereinafter) was used to calculate the gene expression bias index (GEBI) that quantifies the bias for gene expression measures for a group of genes between two conditions and test their statistical relevance by comparing them with GEBIs calculated after randomly rearranging these measures (Fig. S2) (*Díaz-Castillo, 2015*). MCW test is particularly suited to study the existence of an extensive bias for a given quantitative measure between two conditions because it is sensitive to the number of elements within the studied dataset with biased measures and the extent of such biases, but also because it considers all the elements in the studied dataset instead of only those that are deemed significant in virtue of arbitrary thresholds (*Díaz-Castillo, 2015*). For this particular case, MCW test was used to confirm the existence of gene expression heterogeneity biases between each one of the post-amputational timepoints (TX) and the day of the amputation (T0), TX *vs* T0 pairwise comparisons. cvGEBIs, indexes calculated using transcript abundance CV, range from 1 if for all probesets under study transcript abundance CV was larger post-amputationally

than in the day of the amputation ($CV_{TX} > CV_{T0}$), and $-1$ if for all probesets under study transcript abundance CV was lower post-amputationally than in the day of the amputation ($CV_{TX} < CV_{T0}$). To ascertain if observed gene expression heterogeneity biases for each TX *vs* T0 comparison were significantly different from those expected for pairwise comparisons with the same number of elements and value distribution just by chance, cvGEBIs were recalculated after randomly rearranging transcript abundance CV 10,000 times with no restriction. Since later on MCW tests with different designs will be used to address different proximate questions, the design with unrestricted permutation of gene expression measures will be referred to as unrestricted MCW tests.

Blastemas typically form 6–12 days after amputation depending on the animal size and age (*Voss et al., 2015*). If the blastema formation indeed encompassed intervening cell dedifferentiation, and this concurred with an extensive relaxation of chromatin, we would expect that cvGEBIs became negative and significantly different from those expected by chance from day 6–12 onwards. Somehow surprisingly, not only cvGEBIs were significantly different for the expected timepoints, but they started being significantly negative as early as 1.5 days after amputation (Fig. 1, and Table S2).

An intriguing aspect of gene expression heterogeneity dynamics along the regenerative development of axolotl forearm is that it is considerably variable itself (Fig. 1, and Table S2). Interestingly, cvGEBI variation is reminiscent of the progression of axolotl forearm regenerative development stages defined morphologically (*Voss et al., 2015*). In general, while significantly negative cvGEBIs tend to be higher for timepoints that correspond to the transition between regenerative development stages, and lower for intermediate timepoints for each stage (Fig. 1, and Table S2). For example, an increase in cvGEBIs can be observed in the transition between pre-bud and early bud stages, or medium and late bud stages, whereas the lowest cvGEBIs correspond to intermediate timepoints for medium bud and pallet stages (Fig. 1, and Table S2).

Voss and coworkers documented that samples corresponding to the same timepoint can be classified into consecutive morphological stages, and that this staging imprecision is more accentuated for timepoints corresponding to the transition between stages than for intermediate timepoints within each stage (*Voss et al., 2015*). An accentuated variation in the developmental progression of axolotl regenerative development is well known even under highly controlled conditions (*Tank, Carlson & Connelly, 1976*). Thus, the parallelism observed for cvGEBIs dynamics here and morphological changes in the original study for axolotl forearm regenerative development suggests that the methodology used here to analyze gene expression heterogeneity dynamics is sensitive enough to detect the effect of intrinsic factors contributing to gene expression heterogeneity such as axolotl regenerative development inherent asynchrony. Most importantly, despite the potential contribution of regenerative development asynchrony to gene expression heterogeneity, post-amputational cvGEBIs are significantly negative 1.5 days post-amputation onwards, as it would be expected if samples encompassing these timepoints were populated by dedifferentiated cells with extensively relaxed chromatin.

Whether the significant reduction in gene expression heterogeneity here reported truly reflected the presence of a significant number of dedifferentiated cells with extensively
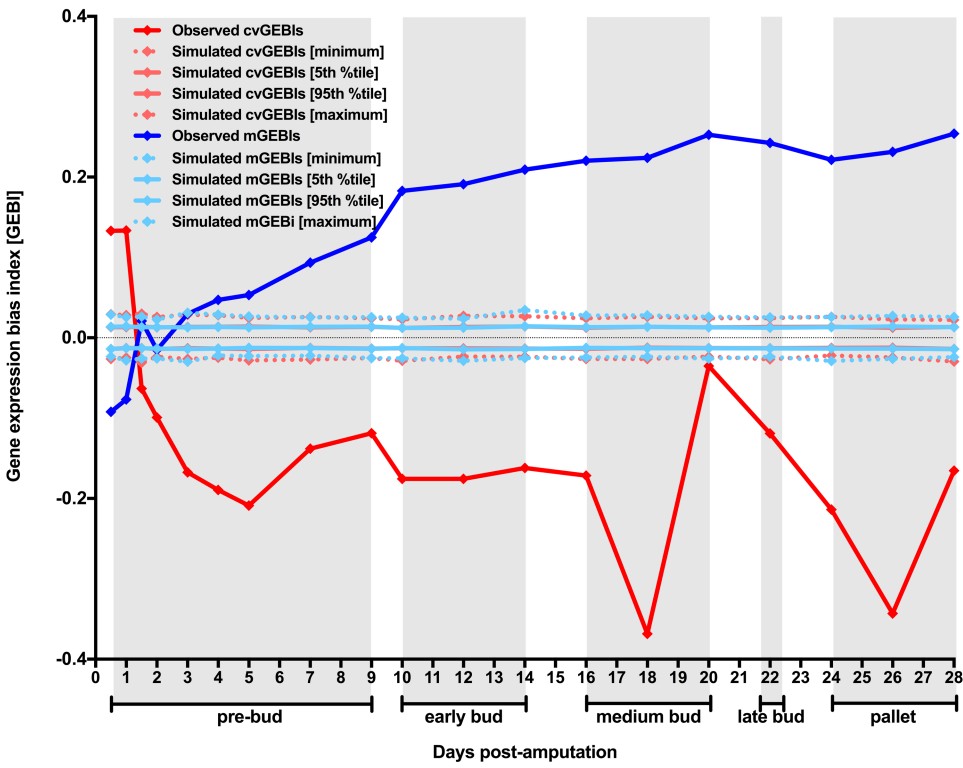

**Figure 1** **Temporal dynamics of the level and the heterogeneity in gene expression along axolotl fore-arm regenerative development.** Generalized changes in gene expression measures along axolotl fore-arm regeneration was inspected using data originally produced by Voss and coworkers and unrestricted MCW tests (see Material and Methods for further details). Gene expression bias indexes (GEBIs) measure generalized biases for transcript abundance CV and mean (cvGEBIs and mGEBIs respectively) when comparing data for each post-amputation timepoint (TX) and the day of the amputation (T0), TX *vs* T0 comparisons. Positive GEBIs represent timepoints for which gene expression measures tend to be higher post-amputationally than in the day of amputation (TX > T0), whereas negative GEBIs represent cases for which gene expression measures tend to be lower post-amputationally than in the day of amputation (TX < T0). Simulated GEBIs were obtained after randomly rearranging gene expression measures for all probesets within TX *vs* T0 comparison 10,000 times with no restriction. The distribution of simulated GEBIs is summarized using minimums, 5th and 95th percentiles, and maximums. Observed GEBIs were considered significant if they outlied the area of the graph defined by 5th and 95th percentiles ($P < 0.05$). Grey boxes delimit axolotl forearm regeneration stages defined by morphological changes according to *Voss et al. (2015)*.

relaxed chromatin within post-amputational forearm blastemas would require independent corroboration using methodologies that go beyond the scope of this article. However, further analyses of the dataset under study resulted in other observations that favor this hypothesis over other obvious explanations for the observed decline in gene expression heterogeneity.

## Gene expression heterogeneity dynamics along axolotl forearm regenerative development cannot be explained by a generalized increase in gene expression

In principle, the decrease in gene expression heterogeneity here detected for axolotl forelimb regenerative development could be explained without assuming that intervening cells undergo extensive chromatin relaxation. It has been shown that cells participating in regenerative processes tend to show a generalized increase in gene expression (*Percharde, Bulut-Karslioglu & Ramalho-Santos, 2017*). Since, gene expression heterogeneity is known to negatively correlate with gene expression level (*Liu, Francois & Capp, 2016*; *Symmons & Raj, 2016*), it would be expected that blastemal samples showed a reduction in gene expression heterogeneity regardless of intervening cells undergoing extensive chromatin relaxation.

To directly test whether the reduction in gene expression heterogeneity here detected resulted from blastemal cells generalized increase in gene expression, first, I proceed to study the dynamics of the level of gene expression for all genes in the dataset under study. Unrestricted MCW tests using transcript abundance mean per probeset and timepoint as proxy for their level of gene expression were used to quantify gene expression level biases across the transcriptome for TX *vs* T0 comparisons. GEBIs calculated using transcript abundance means are referred to as mGEBIs. Consistent with the possibility that blastemal cell showed a generalized increase in gene expression, mGEBIs become positive and significantly different from mGEBIs expected by chance 2 days after amputation (Fig. 1, and Table S2). In other words, genes tend to become significantly overexpressed post-amputationally starting 2 days after amputation. Furthermore, at a first glance, cvGEBI and mGEBI temporal dynamics are quasi-specular, underscoring the possibility that gene expression heterogeneity decrease along axolotl forearm regenerative development were just a consequence of blastemal cell generalized increase in gene expression. However, mGEBIs become significantly positive 2 days after amputation, with a small delay with regard to cvGEBIs becoming significantly negative 1.5 days after amputation. Such delay is more consistent with the possibility that blastemal cell generalized increase in gene expression was a consequence of intervening cells extensive chromatin relaxation that might contribute to the earlier significant decrease in gene expression heterogeneity, than the latter being a side consequence of blastemal cell generalized increase in gene expression.

To further study the interrelationship of the dynamics for gene expression level and heterogeneity along axolotl forearm regenerative development, three different variations of MCW tests for transcript abundance CV and TX *vs* T0 pairwise comparisons were used (Fig. S2). Unrestricted MCW tests were used to ascertain if observed biases for transcript abundance CV were significantly different from those biases expected just by chance. Timepoint-restricted MCW tests were used to ascertain if observed biases for transcript abundance CV were significantly different from those expected if gene expression heterogeneity was exclusively dependent on factors acting on the transcriptome as a whole. Finally, expression-restricted MCW tests were used to ascertain if observed biases for transcript abundance CV were significantly different from those expected if gene expression heterogeneity was exclusively dependent on the variation in the mean level of expression

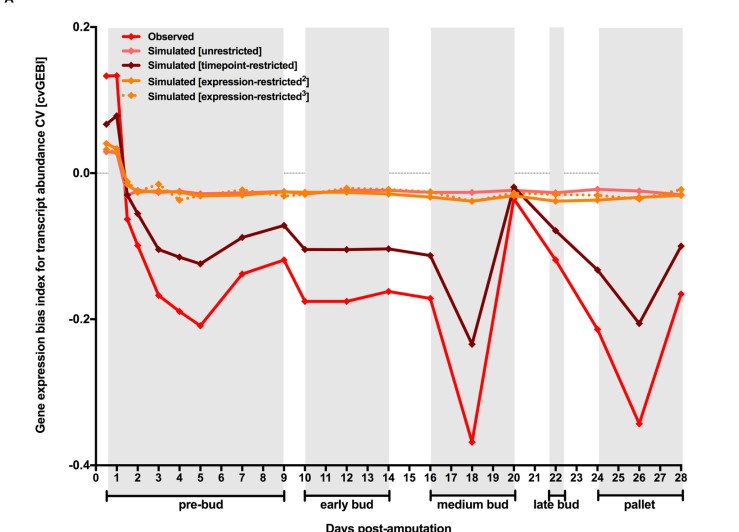
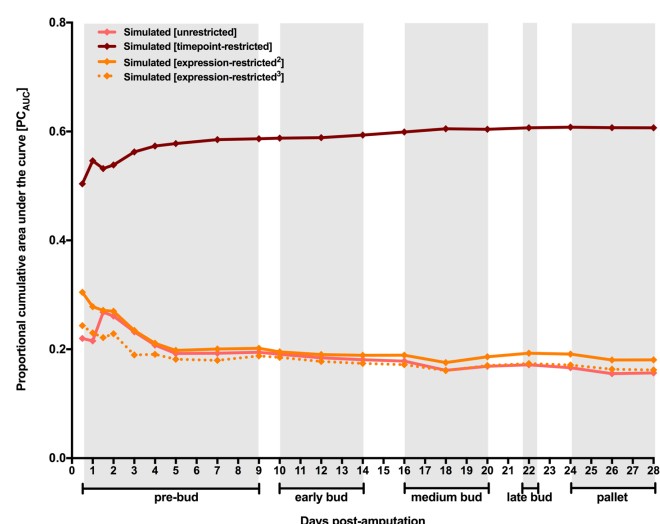

**Figure 2** **Contribution of chance, factors acting on the transcriptome as a whole, and the variation in gene expression level to gene expression heterogeneity dynamics along axolotl forearm regenerative development.** MCW tests using transcript abundance CV for each TX *vs* T0 comparison were performed by randomly rearranging transcript abundance CV with no restriction (unrestricted), restricted by timepoint (timepoint-restricted), or restricted within probeset bins defined according to their transcript abundance mean values (expression-restricted). Two alternative expression-restricted MCW test designs performed independently are indicated with different superscripts (see Material and Methods for further description). MCW tests were repeated 10,000 times, and simulated cvGEBIs with closer values to observed cvGEBIs are represented in (A). (B) The fraction of observed cvGEBI dynamics that could be explained by chance, or factors acting on the transcriptome as a whole, or the variation in gene expression level were calculated as the proportional cumulative area under the curve ($PC_{AUC}$).

for each probeset. If post-amputational gene expression heterogeneity dynamics was just a mere reflection of blastemal cell generalized increase in gene expression it would be expected that observed cvGEBIs were not significantly distinguishable from simulated GEBIs for expression-restricted MCW tests. On the contrary, if post-amputational gene expression heterogeneity dynamics was dependent on factors acting on the whole transcriptome, it would be expected that observed cvGEBIs were not significantly distinguishable from simulated GEBIs for timepoint-restricted MCW tests.

Observed cvGEBIs were significantly different from simulated ones for all timepoints using any of the three MCW test designs (Fig. 2A, and Table S2), suggesting that chance, factors acting on the transcriptome as a whole, or the variation in the level of expression cannot explain on their own gene expression heterogeneity dynamics along axolotl forearm regenerative development. The proportional cumulative area under the curve ($PC_{AUC}$) using simulated cvGEBIs with the closest value to observed cvGEBIs for each timepoint and MCW test design was calculated to estimate how much of gene expression heterogeneity dynamics could be explained by chance, factors acting on the transcriptome as a whole, or the variation in the level of expression (Fig. 2B, and Table S4). The fraction of observed cvGEBIs dynamics that could be explained by the variation in the mean level of expression of each probeset under study is barely higher than what it could be explained just by chance alone (18% *versus* 16%). In stark contrast, factors acting on the transcriptome as a whole could explain up to 60% of observed cvGEBIs temporal dynamics. Considering
the multidimensionality of the dataset, the potential interdependencies that might exist between its elements, i.e., genes that regulate other genes, or the effect of other factors that contributed to gene expression heterogeneity like developmental asynchrony, the fact that simulated cvGEBIs obtained using timepoint-restricted MCWs could explain 60% of observed gene expression heterogeneity dynamics is remarkable. These results strongly suggest that gene expression heterogeneity dynamics along axolotl forearm regenerative development cannot be easily attributed to blastemal cell generalized increase in gene expression. Instead, these results are more consistent with the possibility that axolotl forearm regenerative development concurred with an extensive decrease in gene expression heterogeneity associated to intervening cells undergoing extensive chromatin relaxation, which could be conducive for the expression increase of many genes throughout the genome.

### Gene expression heterogeneity decrease for axolotl forearm regenerative development is consistent with blastemal cell dedifferentiation

The decrease in gene expression heterogeneity for axolotl forearm regenerative development here detected could be explained even if no intervening cell dedifferentiated. Whichever is the origin of blastemal cells, they are expected to encompass a more homogenous pool of cell types than those represented in samples taken at the moment of the amputation, and, immediately after that, when a diverse number of processes are elicited to heal the wound (*McCusker, Bryant & Gardiner, 2015a*). If the decline in gene expression heterogeneity here documented was caused by the relative enrichment of cells from a reduced number of cell types, it would be expected that genes supporting the earliest decline in gene expression heterogeneity resembled the transcriptional profile typical of the few cell types there represented.

To ascertain whether the extensive decrease in gene expression heterogeneity here detected was dependent on the relative enrichment in one or few cell types, I proceed to functionally characterized those probesets that better resemble the sharp decrease in gene expression heterogeneity between 1 and 1.5 days post-amputation (Fig. 1, and Table S2). Probesets in the dataset were ranked with regard to transcript abundance CV fold change for 1 and 1.5 days post-amputation timepoints. The genes corresponding to this prioritized list of probesets and 10 more lists of genes generated by randomly permutating probesets in the dataset with no other restriction were used to identify significant Gene Ontology (GO) enrichments using GOrilla and REVIGO (*Eden et al., 2009*; *Supek et al., 2011*). Only GO terms "mitotic cell cycle process" (GO:1903047) and "chromatin organization" (GO:0006325) appeared significantly enriched for probesets with a decrease in gene expression heterogeneity between 1 and 1.5 days post-amputation (Table 1, and S5). Also, just 10 genes were found supporting the enrichment of both terms out of the 155 genes that support either enrichment, suggesting that the sets of genes supporting each GO enrichment are largely different.

To further characterize the temporal dynamics of transcripts associated to "mitotic cell cycle process" and "chromatin organization" GO terms with regard to the whole

**Table 1  Gene Ontology (GO) enrichment for genes prioritized with regard to their change in gene expression heterogeneity between 1 and 1.5 days after axolotl forearm amputation.** Probesets in *Voss et al. (2015)* dataset were prioritized upon their transcript abundance CV difference between 1 and 1.5 post-amputation timepoints, and randomly rearranged 10 times. GOrilla and REVIGO were used to perform gene ontology (GO) enrichment analyses on prioritized and random lists. The GO term enrichment found using random lists supported by the largest number of genes was used as a threshold to narrow down observed GO enrichments with potential biological significance. See Materials and Methods for further details, and Table S3 for the complete list of GO term enrichments found for prioritized and random lists. Bold names represent genes found supporting both GO term enrichments.

| GO term | Description | *P* value | Enrichment | Genes |
|---------|-------------|-----------|------------|-------|
| GO:1903047 | mitotic cell cycle process | 0.00014 | 1.54 | [*SEPT7, KLHL42, CDK5RAP2, CHAMP1, PAPD5, GSPT1, ORC4, CD2AP, SPDL1, MUS81,* **GSG2***, KIF2A, MAP4, PIBF1, CNOT11, PPM1D,* **USP16***, SPAST, ESPL1, ID2, PSMB3, CDCA5, EIF4E, PSMA5,* **PBRM1***, SMC5, EML4, PSMB6, CHMP5, PSMC5, MAP10, RAD17, CACUL1, PSMC1, NOLC1, FBXW11, CEP152, DYNLT3, PSMD7, INTS3, LMNA, SMC4, PSMD2, PSMD12, CEP250, PSMD11, CUL5, MSH2, RAN, PSMD13,* **TAF10***, FBXL15, NIPBL, TUBA1A, MYH10, MIS12, CCNG2, CEP70, MAP9, CCNG1, VPS4B, MYBL2, TOP2B, TFDP2, TOP2A, CLTA, TUBB4B,* **HMGA2***, RPS6KB1, CDK6, DYNC1H1, RBBP8, DYNC1I2,* **BABAM1***, RAB11A, CDK7, DNM2, OFD1, STAG2, CEP192, NDC80, NDC1,* **NR3C1***,* **HELLS***,* **RUVBL1***, CSNK1A1,* **CCNB1***, KIF23, NUDC, CDCA2, HMMR, NUP153, ZNF207*] |
| GO:0006325 | chromatin organization | 0.00025 | 1.61 | [*ZMYND8, SMARCAD1, VPS72,* **GSG2***, ARID5B, TLK1, BAHD1, NCOR1, PRMT6, KDM5B, SUPT5H, SIRT5, MTA1,* **USP16***, BPTF, DOT1L, KAT2A, BRD8, CHD2, USP7, UTP3, WHSC1,* **PBRM1***, TTF1, PER1, TET3, SCMH1, MTA2, NUCKS1, CREBBP, MORF4L1, CENPP, WDR82,* **TAF10***, MTF2, TAF5, SUPT16H, INO80C, SUPT7L, DNAJC2, HIRA, NFE2, SMARCE1, BRD2, ATRX, RNF2, ATXN7L3, BAZ1A, BAZ1B,* **HMGA2***, PHF20, SMARCC1, PRDM2,* **BABAM1***, SRPK1, OGT, ARID4A, TET2, HIRIP3, CBX3, SMARCA2,* **NR3C1***,* **HELLS***,* **RUVBL1***, SETD2, TLK2,* **CCNB1***, RBM14, KAT6B, PRKCD, ZNF462, ELP3*] |

transcriptome, I used a new variation of the MCW test. Functionally-restricted MCW tests proceed by calculating cvGEBI and mGEBI for probesets associated to genes classified in "mitotic cell cycle process" and "chromatin organization" GO terms before and after randomly rearranging functional subset tags for the whole dataset, respectively (Fig. S2). For both GO terms, the expected sharp decrease in transcript abundance CV between 1 and 1.5 days post-amputation is followed by a sharp increase in transcript abundance mean between 2 and 3 days post-amputation (Fig. 3, and Table S2). Consistent with these trends, the original characterization of the Voss dataset showed significant enrichments for GO terms related with cell cycle and chromosome organization for probesets with a statistically significant difference in transcript abundance between 2 and 3 days post-amputation timepoints (*Voss et al., 2015*).

"Mitotic cell cycle process" and "chromatin organization" GO term enrichment for probesets better supporting the sharp decrease in gene expression heterogeneity between 1 and 1.5 days post-amputation timepoints, and the transcriptomic dynamics of probesets for

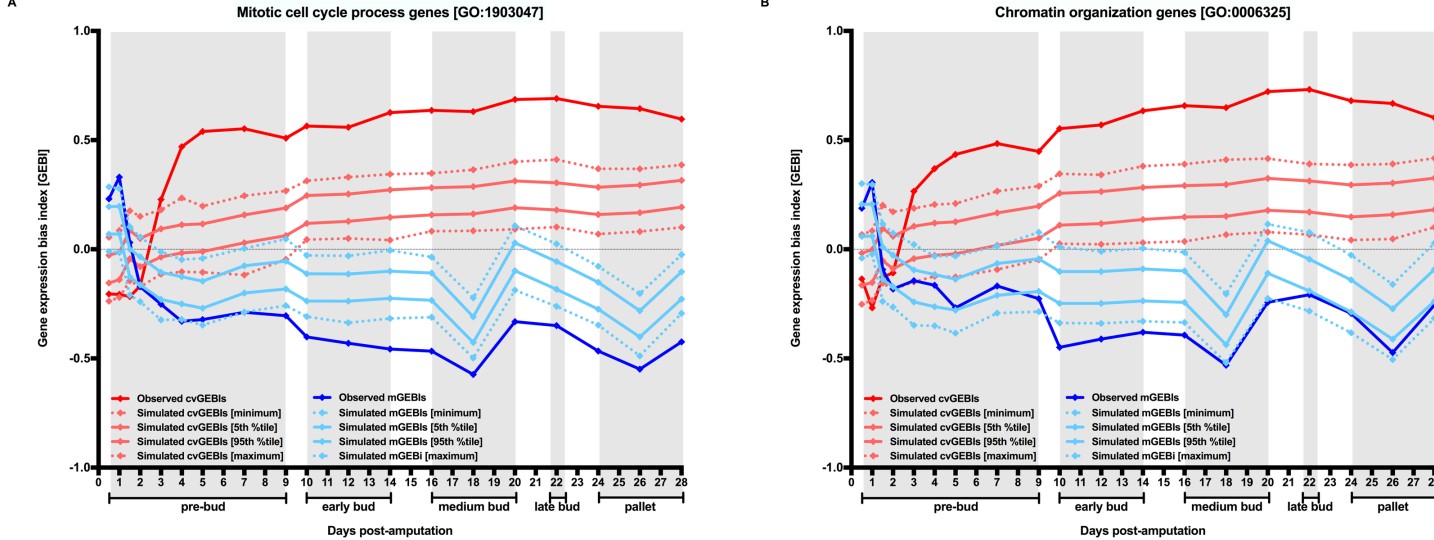

**Figure 3** **Temporal dynamics of the level and the heterogeneity in gene expression along axolotl forearm regenerative development for groups of genes defined by their functionality.** The Gene Ontology Consortium database was used to retrieved genes belonging to ''mitotic cell cycle process'' (GO:1903047) and ''chromatin organization'' (GO:0006325) GO terms (*The Gene Ontology Consortium, 2015*). The temporal dynamics of the level and heterogeneity in gene expression for probesets associated to ''mitotic cell cycle process'' and ''chromatin organization'' genes with regard to the whole transcriptome were inspected using functionally-restricted MCW tests (A and B, respectively). cvGEBI and mGEBI for each group of probesets and TX vs T0 comparison were calculated before and after randomly rearranging functional subset tags for the whole transcriptome 10,000 times. The distribution of simulated GEBIs is summarized using minimums, 5th and 95th percentiles, and maximums. Observed GEBIs were considered significant if they outlied the area of the graph defined by 5th and 95th percentiles ($P < 0.05$). Grey boxes delimit axolotl forearm regeneration stages defined by morphological changes according to *Voss et al. (2015)*.

genes associated to these two terms along axolotl forearm regenerative development does not seem to be consistent with the possibility that early stages of regenerative development coursed with a relative enrichment in one or few cell types. Instead, these trends seem to relate better with the possibility that intervening cells undergo the earliest steps of transformations expected for dedifferentiating cells, namely re-entry in cell cycle and chromatin reorganization. If a reduction in gene expression heterogeneity followed by an increase in the expression of those genes that might drive the process of cell dedifferentiation reflected that chromosome regions encompassing these genes were the first to undergo chromatin relaxation for cells leading regenerative responses suggests that studies focused on the dynamics of the chromatin context for those genes that show the earliest decline in gene expression heterogeneity along regenerative development process could be of great help to better understanding how regenerative development is elicited in response to injury.

## DISCUSSION

In the present article, using already available transcriptomic data for axolotl regenerative development and the assumption that such regenerative processes concurs with cell dedifferentiation, very preliminary support is given to the proposal that cell dedifferentiation concurs with an extensive decrease in gene expression heterogeneity

(*Díaz-Castillo, 2017b*; *Voss et al., 2015*). Although here I used a dataset that has already been characterized, the analyses here presented differ from the original characterization of the dataset mostly because they focus on the study of changes in gene expression heterogeneity; changes to the mean level of expression are used only accessorily. Another important difference of the present study from the original characterization and other transcriptomic studies consist in the use of MCW tests, which permits characterizing extensive transcriptomic trends taking into consideration all the genes encompassed by large datasets, instead of just those showing differences beyond thresholds chosen arbitrarily.

An important caveat to the analyses here presented relates with the increasing notion that commonly used analytical methodologies to study transcriptome dynamics might not fairly represent cases in which the transcriptional output of all or most genes in the genome is concertedly increased (*Percharde, Bulut-Karslioglu & Ramalho-Santos, 2017*). The use of normalization methods based on the assumption that the expression of "housekeeping" genes or the general transcription output of the nucleus is largely invariable between the assayed conditions, and the difficulty in precisely controlling the number of cells contained in samples for complex and very dynamic processes such as regenerative development represent the main difficulties to ensure a fair consideration of cell transcriptional output for transcriptomic analyses (*Percharde, Bulut-Karslioglu & Ramalho-Santos, 2017*). Future analyses, probably focused on single cells, will be needed to assess the impact of this methodological caveat for the study of transcriptome dynamics in general, and for these circumstances that concur with a generalized increase in gene expression in particular.

How many and which, if any, cells dedifferentiate upon injury along regenerative development processes, and if such dedifferentiation indeed concur with extensive chromatin relaxation driving to an extensive decrease in gene expression heterogeneity require independent confirmation. The possibility that changes in gene expression heterogeneity could approximate changes in chromatin configuration for dedifferentiated cells might be of wide interest for biomedical and biotechnological areas based on cell dedifferentiation such as regenerative development, cancer, human iPSCs, or plant somatic embryogenesis. Here, I proceed to briefly discuss how the study of gene expression heterogeneity dynamics might help developing methodologies to monitor and/or control cell dedifferentiation-based processes, and how a transitional reduction in gene expression heterogeneity for dedifferentiating cells can promote undesired longer-term phenotypic heterogeneity.

### The study of gene expression heterogeneity can be useful for the characterization of processes that concur with cell dedifferentiation, and to engineer approaches for their monitoring and modulation

Whole transcriptome analysis is now a common tool to identify relevant changes in transcript abundance reflecting natural or experimental cues. Often, these analyses are based on the dubious assumption that significant changes in transcript abundance directly reflect equally significant changes in the function/effect of the very transcript or the protein it codes for. However, the study of transcript abundance dynamics permits

directly addressing only a small fraction of the regulatory mechanisms needed for the generation of active genetic products. Transcript spatial distribution, and translation, or protein post-translational modification, folding, spatial distribution, and coordinated functionality with other genetic products remain largely unapproachable using only transcriptomic analyses. Since all these steps in the *de novo* synthesis of active genetic products are susceptible to be regulated independently from regulatory mechanisms modulating transcript abundance, it is possible that very obvious changes in transcript abundance are phenotypically insignificant because they are buffered by the system, or genes for which no significant change is detected at a transcriptional level had key roles in certain processes because of regulatory inputs acting at the protein level. This limitation is particularly troublesome when transcriptomic analyses are used to distinguish genes that drive a particular process from those that reflect the progression of the process in question, i.e., drivers *versus* effectors. While such distinction might be meaningless for the identification of diagnostic markers for biological processes, it is key for the basic characterization of such processes and the efficient engineering of strategies that afford *in vivo* monitoring or modulating their progress.

Gene expression heterogeneity is not directly informative about the abundance of active genetic products, and, therefore, its use to infer functional changes in the process under consideration is also very limited. However, gene expression heterogeneity dynamics might be highly informative about regulatory mechanisms acting on chromatin accessibility, transcription, and transcript degradation. In particular, the integration of gene expression heterogeneity dynamics with proxies for higher order chromatin organization could be of great value to identify chromosome regions encompassing genes that undergo the earliest changes along naturally-occurring or induced cell dedifferentiation, and the characterization of the regulatory elements mediating such changes. Furthermore, the possibility of identifying chromosome regions showing the earliest changes associated to cell dedifferentiation would afford using them as target for the insertion of engineered genetic constructs that help better monitoring or even modulating processes that concur with cell dedifferentiation. For example, in the context of regenerative development in response to injury, chromosome regions undergoing the earliest chromatin relaxation along cell dedifferentiation could be targeted for the insertion of genetic constructs coding for visible markers that helped identifying which cells within a complex tissue are susceptible to dedifferentiate in response to injury, or monitoring the progress of regenerative responses in non-invasive ways. Alternatively, the same regions could be targeted for the insertion of constructs coding for RNAs of proteins that by stimulating or interfering with key elements of cell dedifferentiation-based processes permitted modulating the course of such processes and improve their efficiency or minimize detrimental side effects.

## A transitional reduction in gene expression heterogeneity for dedifferentiating cells could promote a long-term phenotypic heterogeneity

A troublesome aspect of processes that concur with cell dedifferentiation is the accentuated heterogeneity detected at genetic, epigenetic, and phenotypic levels

(*Almendro, Marusyk & Polyak, 2013*; *Burrell et al., 2013*; *Fossati, Jain & Sevilla, 2016*; *Krishna et al., 2016*; *Ling et al., 2015*; *Meacham & Morrison, 2013*). Such heterogeneities are of great concern because they can limit the response to treatment of cancerous cell, or the reproducibility of healthy clonal tissues and individuals derived from animal and plant induced dedifferentiated cells (*Almendro, Marusyk & Polyak, 2013*; *Burrell et al., 2013*; *Fossati, Jain & Sevilla, 2016*; *Krishna et al., 2016*; *Ling et al., 2015*; *Meacham & Morrison, 2013*). Intriguingly, a chromatin relaxation-based extensive reduction in gene expression heterogeneity for dedifferentiated cells could promote long-term phenotypic heterogeneities following cell dedifferentiation by increasing genetic and epigenetic mutagenic potential and the phenotypic relevance of preexisting or newly generated mutations for dedifferentiated cells.

On one hand, an extensive chromatin relaxation-based surge of DNA accessibility might result in a larger number of coding or regulatory loci being susceptible to insults that result in genetic mutations, but also the activation of genetic elements that can directly intervene in mutagenic events such as those transposable elements populating regions of highly compacted chromatin. Indeed, abundant reports exist for accentuated genetic variation, and activities of transposable elements in processes concurring with cell dedifferentiation (*Cooper et al., 2017*; *Jiang et al., 2011*; *Ling et al., 2015*; *Macia, Blanco-Jimenez & Garcia-Perez, 2015*; *Wang & Wang, 2012*; *Zhu et al., 2012a*). Furthermore, the increased activity of repair machineries that help protecting the integrity of the genome in dedifferentiated cells might evidence their enhanced potential for genetic mutagenesis (*Cooper et al., 2017*). Also, 5-methylcytosine (5mC) is highly prone to spontaneously deaminate into thymine (T), which if uncorrected would cause both genetic and epigenetic mutations (*Bellacosa & Drohat, 2015*; *Cortazar et al., 2007*). The restoration of 5-methylcytosine is not straightforward and requires a concatenation of enzymes that recognize thymine:guanidine (T:G) mismatches, excise T, restore the original cytosine (C), and methylate the restored C (*Bellacosa & Drohat, 2015*; *Cortazar et al., 2007*). The key enzyme for 5mC deamination repair is thymine glycosylase (TDG), which is known to be considerably inefficient (*Bellacosa & Drohat, 2015*; *Cortazar et al., 2007*). The accentuated tendency to deaminate of 5mC, the complicated way of restoring 5mC, and the inefficiency of some of the elements required for 5mC restoration suggest that for any given methylated locus in a population of isogenic cells in the same environment there would be a variable assortment of cells showing 5mC, T, or C. Since the limiting step for 5mC deamination is DNA melting (*Fryxell & Moon, 2005*), processes more permissive for DNA melting, such as dedifferentiated cells extensive chromatin relaxation, could result in an accentuated and ultimately stochastic variation in DNA methylation and 5mC deamination-based mutations.

On the other hand, beyond the direct effect chromatin relaxation in dedifferentiated cells might have on the genetic and epigenetic mutagenic potential, a transitional chromatin relaxation-dependent extensive decrease in gene expression heterogeneity can also promote a long-term phenotypic heterogeneity by enhancing the phenotypic relevance of pre-existing and newly generated genetic and epigenetic mutations. Genetic capacitance refers to the accumulation and release of genetic variation in a cryptic state, i.e., not causing phenotypic variation (*Gibson & Reed, 2008*; *Masel & Trotter, 2010*;

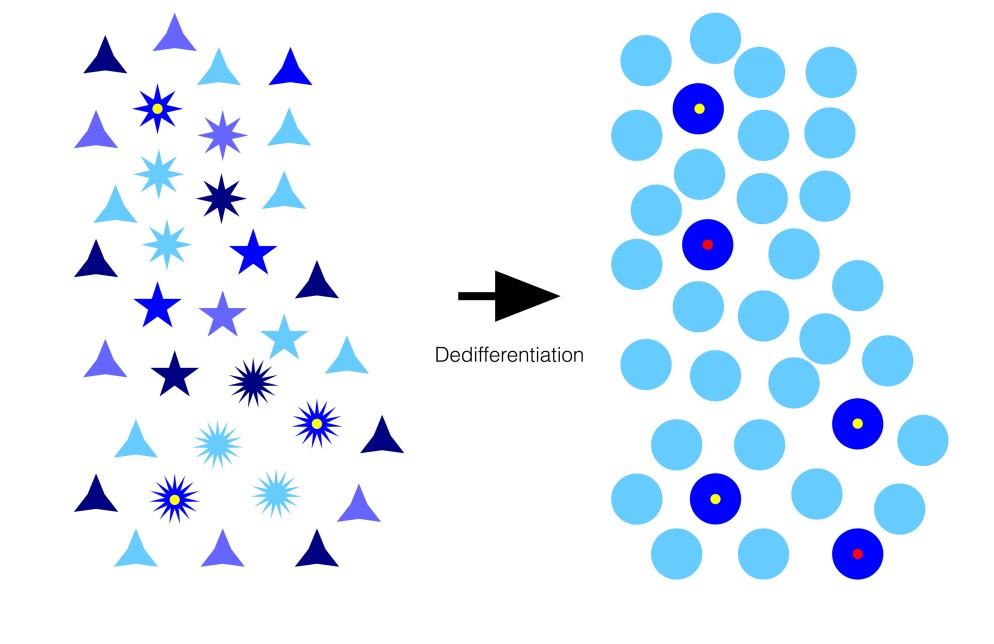

★ ✳ ✱ ▲ Differentiated cell types  ● Dedifferentiated cell type  ● Pre-existing genetic mutation  ● New genetic mutation

**Figure 4** **Generalized reduction in gene expression heterogeneity upon cell dedifferentiation might reduce genetic capacitance.** Cartoon symbolizing how genetic variation can become phenotypically relevant associated to the chromatin relaxation-dependent reduction in gene expression heterogeneity upon cell dedifferentiation. Blue shades represent the variation in gene expression for a particular gene in a population of cells before and after dedifferentiation. Pre-existing or new genetic mutations might be phenotypically more distinguishable upon cell dedifferentiation because of the generalized reduction in gene expression heterogeneity that cause a narrowing of the spectrum of stochastic phenotypes.

*Paaby & Rockman, 2014*). In recent years, it has been suggested that chromatin compaction-dependent gene expression heterogeneity promotes genetic capacitance because it results in a spectrum of stochastic phenotypes from which some genetic-based phenotypes might be indistinguishable (*Díaz-Castillo, 2015*; *Díaz-Castillo, 2017a*). Such cryptic genetic variation would be undetectable for selective forces, and, therefore allowed to fluctuate randomly within biological populations. Generalized differences in chromatin compaction modulating gene expression heterogeneity, and, with it, genetic capacitance, have been proposed to importantly contribute to sexually dimorphic traits in metazoans and differences in the spatiotemporal dynamics of natural populations from species with different amounts of junk DNA (*Díaz-Castillo, 2015*; *Díaz-Castillo, 2017a*).

If cell dedifferentiation truly concurred with an extensive decrease in chromatin compaction-dependent gene expression heterogeneity, it could be expected that such decrease resulted in a reduction in genetic capacitance (Fig. 4). In other words, due to the extensive decrease in gene expression heterogeneity, the spectrum of stochastic phenotypes would become narrower, and, consequently, preexisting cryptic or new genetic and epigenetic mutations would become phenotypically relevant (Fig. 3). Revealed genetic and epigenetic mutations would contribute to a phenotypic plasticity in dedifferentiated

cells noticeable in the absence of any other cue, or upon signaling from their environments or the action of external treatments. The growing literature on the relevance of stochastic events commonly occurring in early development could support the possibility that un/dedifferentiated cells showed limited genetic capacitance because, at least in part, their extensive chromatin relaxation, and, therefore, being more susceptible to manifest plasticity inherently and/or in response to external cues (*Losick & Desplan, 2008*; *Vogt, 2015*).

In summary, an extensive reduction in gene expression heterogeneity for dedifferentiated cells is not only not conflicting with the multiple reports on accentuated genetic, epigenetic, and phenotypic heterogeneities for natural or induced processes that concur with cell dedifferentiation, but it could be argued that the extensive chromatin relaxation typical of dedifferentiated cells might be the ultimate cause for these apparently opposing heterogeneity trends.

## CONCLUDING REMARKS

The high precision and resolution achieved with modern techniques is providing us with very large collections of data for all sorts of biological systems and processes. Less progress has been made analytically to squeeze as much biologically relevant information as possible from such highly multidimensional datasets. Although the study of biological variation observed in the absence of genetic and environmental cues and its usefulness for basic and applied biology is still very much underrated, an appreciation for the study of mechanisms that cause biological variation and its potential for biomedical applications is starting to grow (*Rosenberg & Queitsch, 2014*). That cell dedifferentiation might concur with an extensive reduction in gene expression heterogeneity might be of great interest for biomedical and biotechnological processes that depend on cell dedifferentiation, i.e., regenerative development, cancer, human iPSCs, or plant somatic embryogenesis. The study of cell dedifferentiation-based gene expression heterogeneity dynamics opens a complementary way to characterize these processes, especially in their earliest stages, and lay the foundation to newer, more precise tools that help monitoring and modulating them. Furthermore, the possibility that a reduction in gene expression heterogeneity can promote long-term phenotypic heterogeneity following cell dedifferentiation points to the possibility of identifying key elements to target in an attempt to minimize the adverse effect such phenotypic heterogeneity might have for the response to treatment, or the reproducibility of healthy clonal tissues and individuals. More generally, the utilization of appropriate analytical methods to characterize gene expression heterogeneity dynamics as an adjunct to already existing methods might help accessing layers of information buried in today's multidimensional datasets and offer a more representative understanding of biological systems and processes.

## ACKNOWLEDGEMENTS

I want to thank Min Zhao, and three anonymous reviewers for their constructive criticism during the revision of this article. I want to thank Randal Voss and Zeba Wunderlich for valuable comments during the preparation of this article. I want to express my deepest

gratitude to Sue Bryant, David Gardiner, and Kate McCusker for valuable comments during the preparation of this article, and their generosity and support during my acclimation to the study of axolotl regenerative development. I want to express my eternal gratitude to Raquel Chamorro-Garcia for valuable comments during the preparation of this article, and her unfailing support.

### Funding

The authors received no funding for this work.

### Competing Interests

The authors declare there are no competing interests.

### Author Contributions

- Carlos Díaz-Castillo analyzed the data, wrote the paper, prepared figures and/or tables, reviewed drafts of the paper.

### Data Availability

Raw data analyzed in this manuscript was obtained from Gene Expression Omnibus (GSE67118).

### Supplemental Information

Supplemental information for this article can be found online at http://dx.doi.org/10.7717/peerj.4004#supplemental-information.

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
