# Peer review of "Transcriptome dynamics along axolotl regenerative development are consistent with an extensive reduction in gene expression heterogeneity in dedifferentiated cells"

_PeerJ, doi:10.7717/peerj.4004_

## Round 0.1 · original submission · Major Revisions

· Academic Editor

Major Revisions

As suggested by reviewer 1, the data interpretations are controversial. It will be better to add more solid details and delete some that may be overinterpreted.

Reviewer 1 ·

Basic reporting

The manuscript needs work on sentence structure. Specifically, the overuse of prepositional phrases makes the sentences very difficult to follow. I suggest that the writer limits the use of these phrases and the manuscript will be much more reader friendly.

Experimental design

The analysis seems sound, but the interpretations for what may be occurring within the blastema are overreaching and sometimes based upon speculation. Little is known about the cell heterogeneity across the time points studied and almost nothing is known about the chromatin structure in these cell type. The study is interesting and of value, but toning back the interpretation seems appropriate. The following suggestions point to examples where changes could be implemented.

Validity of the findings

-I am not disputing the possibility that gene expression heterogeneity decreases during dedifferentiation, but it is unclear if the gene expression measures utilized in this study can support this hypothesis. This argument is based upon several points:

1) Our understanding of cell dedifferentiation during limb regeneration is minimal, which is an assumption required for this analysis to be applicable. As far as I am aware, only one cell type, the fibroblast, is assumed to dedifferentiate during axolotl limb regeneration and the definition of this dedifferentiation is unclear. Could fibroblasts just be “activated” fibroblasts similar to myofibroblasts in mammals, yet have more capabilities for patterning. Is the level of dedifferentiated limb fibroblasts known? As far as I know it is not. The fibroblast population in the limb is a small proportion of the cells that make up the sample used for transcriptomic analysis. Could the conclusions made be a representation of gene expression not actually occurring in the dedifferentiated fibroblasts?

2) The sharp decrease in gene expression heterogeneity after injury does not necessarily represent dedifferentiation. For example, amputation leads to a retraction of muscle fibers and their decrease in expression of many muscle-specific genes, which is likely highly heterogeneous in its extent and temporal profile. This phenomenon would lead to high variability early on and decreased variability after the event occurs.

“Somehow surprisingly, not only cvGEBIs were significantly different for the expected timepoints, but they started being significantly negative as early as 1.5 days after amputation (Figure 1, and Table 1). If a reduction in gene expression heterogeneity truly reflected the presence of a significant number of dedifferentiating cells, these results would suggest that cell dedifferentiation leading to the formation of the blastema can be noticed transcriptomically considerably earlier than it can be defined morphologically.”
-This statement assumes that dedifferentiation is the only transcriptional profile being collected, which is not the case. During this time, the inflammatory response, wound closure, and histolysis are all beginning to reside, which should also generate a decrease in transcriptional heterogeneity once they subside. I don’t think it is appropriate to say that the signature is due to dedifferentiation.

4) The interpretations made by studying cell heterogeneity could be innapropriate because the readout is the average signals generated from a mass of heterogeneous tissue including muscle cells, vascular cells, Schwann cells, epidermal cells, skeletal cells, and fibroblasts. Assume that the early blastema contains many different cell types, but may be different from sample to sample. Over time, the majority of cells in the blastema are of a single cell type, likely the fibroblast. This scenario would give the same gene expression heterogeneity profile seen here, but the interpretation would be considerably different than the one presented in the manuscript. Without single cell data, it is difficult to prove that the loss of gene expression heterogeneity demonstrated here is due to cell dedifferentiation.

-Can a sentence be included in the GO analysis stating that the genes represented in the mitotic re-entry category are significantly different than the chromatin organization category. Chromatin organization in terms of GO is often associated with the cell cycle instead of the euchromatic or heterochromatic nature of the chromatin.

-It is argued in the manuscript that chromatin architecture changes globally during regeneration. I am unaware of a study that demonstrates this to be the case. The author argues this point, but it is speculation until a study demonstrates this to be the case.

-Can the author provide evidence that 10 biological replicates is sufficient for testing heterogeneity of gene expression? They present data that lower biological replication is likely insufficient, but can an argument be made that 10 replicates is enough?

-The authors should make clear how the analysis presented here varies from the original Voss et al., 2015 manuscript.

Additional comments

The manuscript by Diaz-Castillo presents some interesting points on the general feature that dedifferentiation is associated with a loss of gene expression heterogeneity. The manuscript uses a deep transcriptional dataset of axolotl limb regeneration to study gene expression heterogeneity throughout the regenerative process. Although the analysis seems sound, the interpretations for what may be occurring within the blastema are overreaching and sometimes based upon speculation. Little is known about the cell heterogeneity across the time points studied and almost nothing is known about the chromatin structure in these cell type. The study is interesting and of value, but toning back the interpretation seems appropriate. The following suggestions point to examples where changes could be implemented.
Most points are covered in the above sections.

Reviewer 2 ·

Basic reporting

NA

Experimental design

This manuscript performs a set of investigations and provides empirical evidence for the hypothesis that dedifferentiated cells will lead to reduced gene expression heterogeneity. Motivation and problem are clearly described.
Methods are described with proper details, I would suggest to add more biological explanation for each metric used in the study.

Validity of the findings

An existing data set is used in the study. The effect of the number of replicates is discussed, highlighting the importance of enough replicates.

Several different metrics (including proper statistical test) are used to analyse the data and derive final conclusions.

Additional comments

The manuscript presents empirical results for the hypothesis of reduced gene expression heterogeneity in dedifferentiated cells. Overall, the paper is well organized and presented. Conclusions are well derived from the analysis. My only suggestion is to add more biological meaning for the metrics used in the study.

---

## Round 0.2 · Minor Revisions

· Academic Editor

Minor Revisions

Please add more detail for the methods and strengthen the logic connections as suggested by reviewer 2.

Reviewer 1 ·

Basic reporting

The author has made significant changes to the manuscript to make it easier to follow.

Experimental design

The experimental design was based upon a previously published study, which was strong.

Validity of the findings

My general concern in my original review was the interpretation of the findings. Many of the issues cannot be resolved without years of research in the field to determine the actual meaning of dedifferentiation and the cellular heterogeneity of the regenerating limb blastema. The Author has made significant changes to provide support for their idea of dedifferentiation and I accept the arguments made by the author.

Additional comments

The author has strengthened the manuscript through the changes made throughout the manuscript and their rebuttal to the reviewers is valid. I am comfortable supporting acceptance of the manuscript.

Reviewer 3 ·

Basic reporting

The sentences are really difficult to understand. A native speaker is suggested to revise the manuscript.

cvGEBIs is the major concept for this paper. But it's not well described in the method. It would be nice to provide a formula or diagram to readers.

Experimental design

The idea of studying changes of expression heterogeneity is very interesting. However, the logic is not complete to get the conclusion.

Validity of the findings

The logic connection between extensive chromatin relaxation and generalized decrease in gene expression heterogeneity along with cell dedifferentiation is not clear. How extensive chromatin relaxation causes generalized decrease in gene expression heterogeneity is not explained in the background.

The normalization method strongly affect the detection of generalized increase in transcription. The author should do more research about it. Other wise it's really difficult to convince people that generalized increase in transcription doesn't contribute to the decreasing of expression heterogeneity.

The fact that genes have sharp decrease in gene expression heterogeneity between 1 and 1.5 days post-amputation are enriched in cell cycle and chromatin organization terms doesn’t suggest these two functional terms are the mechanism of decreasing gene expression heterogeneity. Instead the author should test if genes that are deferentially expressed post-amputation are enriched in these two functional terms.

---

## Round 0.3 · accepted · Accept

· Academic Editor

Accept

It is great to see the uodated manuscript have addressed all the concerns from reviewers.